# Absence of community-acquired *Candida auris* colonization among newly hospitalized participants without recent healthcare exposure from a cross-sectional study in Dhaka, Bangladesh

Gazi Md. Salahuddin Mamun,[1] Tanzir Ahmed Shuvo,[1] Sanzida Khan,[1] Syeda Mah-E-Muneer,[1,2] Md. Aminul Islam,[1] Dilruba Ahmed,[1] Kabid Ahmed,[1] Debashis Sen,[1] Kamal Hossain,[1] Md. Nazmul Islam,[3] Aninda Rahman,[3] Mohammad Monir-Uz-Zaman,[4] Md. Mustafizur Rahman,[4] Fahmida Naz Mustafa,[4] Md. Salim,[4] Rubina Yasmin,[4] Md. Shafiur Rahman,[5] Tarak Nath Kundu,[5] Mostafa Kamal,[5] Farzana Sohael,[5] Sakina Shab Afroz,[5] Mahmudur Rahman,[6] Fahmida Chowdhury[1]

**ABSTRACT** *Candida auris* (*C. auris*), an emerging fungus, is primarily recognized for causing healthcare-associated infections. While carriage among hospitalized patients is well documented, community spread remains poorly understood. This study aimed to determine the burden of community-acquired *C. auris* skin colonization among newly hospitalized participants without any recent healthcare exposure. This cross-sectional study was conducted at two tertiary-level government hospitals and their surrounding catchment areas in Dhaka, Bangladesh, from June 2023 to May 2024. We enrolled 800 participants (400 from each hospital) shortly after admission, meeting the following criteria: no healthcare exposure within the past three months, no invasive procedures performed since the current hospital admission, and residence within the hospital's catchment area for the past three months. Skin swabs from the axillae and groins were cultured on CHROMagar Candida Plus to determine fungal colonization. Presumptive positive isolates were identified using VITEK 2, and *C. auris* was confirmed by whole-genome sequencing. No *C. auris* colonization was detected in the axillary or groin in any enrolled participant. However, 128 (16%) patients were found to be colonized with other fungal species. The absence of community-acquired *C. auris* colonization in this study suggests that transmission primarily occurs within hospital settings. However, our strict inclusion criteria, which excluded patients with recent healthcare exposure, may have influenced this finding. Further research using alternative study designs is needed to fully understand the potential for *C. auris* spread within the community.

**IMPORTANCE** *Candida auris*, a rapidly emerging multidrug-resistant fungal pathogen with high mortality, poses a critical global health threat, particularly in healthcare settings. While extensive research has focused on its colonization among hospitalized patients, its potential presence and community transmission remain largely unexplored. An earlier study among critically ill patients in Bangladesh detected four colonized cases of *C. auris* without hospitalization history within the past year and admitted directly from home (F. Chowdhury et al., unpublished data). Notably, three participants resided in the same district and one in an adjacent district, raising serious concerns about possible community spread. This study is crucial in addressing this knowledge gap by assessing the burden of community-acquired *C. auris* colonization among newly hospitalized patients in Dhaka. Understanding its potential transmission outside the hospital is vital for shaping public health responses, guiding infection control strategies, and strengthening global surveillance efforts to mitigate the spread of this highly resistant pathogen.

**Peer Reviewer** Milena Kordalewska, Hackensack Meridian Health, Nutley, New Jersey, USA

Address correspondence to Tanzir Ahmed Shuvo, tanzir@icddrb.org.

The authors declare no conflict of interest.

See the funding table on p. 8.

10.1128/spectrum.00393-25 **1**

**KEYWORDS** *Candida auris*, emerging fungus, colonization, community-acquired, Bangladesh

Fungal diseases contribute significantly to global morbidity and mortality (1). *Candida* species, commonly present as commensal flora on human skin, are detectable in up to 60% of healthy individuals (2). However, invasive *Candida* infections, including sepsis in critically ill patients, typically arise from colonization combined with local or systemic immune suppression (2, 3).

*Candida auris* (*C. auris*) is an emerging, often multidrug-resistant fungus primarily associated with healthcare settings (4). Studies have shown that about 10% of *C. auris*-colonized individuals develop invasive infections, particularly bloodstream infections, contributing to high mortality rates ranging from 20% to 60% (5). The World Health Organization has designated *C. auris* as a critical priority pathogen due to its potential for outbreaks of invasive candidiasis associated with high mortality rates (6). Moreover, the US Centers for Disease Control and Prevention has classified *C. auris* as an urgent antimicrobial-resistant threat (7).

*C. auris* can colonize the skin for extended periods with prior healthcare exposures (e.g., being on a ventilator, receiving systemic fluconazole or carbapenem antibiotics in the past 90 days, or having ≥1 acute care hospital visit in the past six months) identified as risk factors for colonization (8). Environmental contamination with *C. auris* is also associated with the high burden of skin colonization with *C. auris* among residents (9). Though the molecular mechanisms of skin colonization are still unclear, studies among mice have reported that *C. auris* Hog1 mitogen-activated protein kinase and a *C. auris*-specific adhesin, surface colonization factor (Scf1), are essential for efficient skin colonization and intradermal persistence (10, 11).

Data on community-acquired *C. auris* colonization are limited. One UK study reported a 6.8% colonization rate among intensive care unit (ICU) patients in four academic hospitals (12). While Bangladesh has a high burden of *Candida* spp. infections, particularly superficial mycoses (13), a recent study in Dhaka found 4% *C. auris* colonization among ICU patients within 48 hours of admission and two cases of bloodstream infection (F. Chowdhury et al., unpublished data). Among the *C. auris*-colonized patients, four had no history of hospitalization within the past year and were living in the nearby geographic area. Due to the possibility of community transmission among them, we conducted this study to create better evidence on it. However, due to the short enrollment window and the possibility of prior ward admissions, community acquisition could not be definitively determined.

Though there are several published papers on fungal skin infections, to our best knowledge, no studies have specifically investigated fungal skin colonization among newly hospitalized participants who came directly from the community or household, without recent healthcare exposure. This study aimed to determine the burden of potentially community-acquired *C. auris* in this population.

## MATERIALS AND METHODS

### Study design, duration, and setting

This cross-sectional study was conducted from 11 June 2023 to 30 May 2024 at two government tertiary-level hospitals (site 1 had 500, and site 2 had 1,350 beds) and their surrounding catchment areas of Dhaka, Bangladesh. These hospitals, located 10 kilometers apart, serve patients from all over the country, primarily those residing in the local catchment area.

### Study population

Study participants of all ages and sexes were enrolled from newly admitted patients in the internal medicine, pediatrics, gynecology and obstetrics, and general surgery

departments. Enrollment from these departments aimed to achieve a balanced representation of male, female, and pediatric participants. Participant selection was not influenced by the admitting department. Inclusion criteria were: no overnight hospital admission within the past three months, no invasive procedures since the current hospital admission, and residence within the hospital's catchment area for the past three months. Participants with any prior hospital stay within the last three months were excluded to minimize the possibility of healthcare-associated colonization. Most patients admitted to these hospitals were from a lower or lower-middle socioeconomic background.

To avoid the challenges and high refusal rate anticipated in collecting skin swabs from the axillary and groin in a community household setting, we enrolled the participants shortly after hospital admission to detect community-acquired colonization. If *C. auris* colonization had been detected among newly admitted participants, this approach would have facilitated subsequent investigation of household transmission among their household members.

## Study procedures and data collection

After obtaining written informed consent (or from legal guardians for minors), eligible participants were enrolled as soon as feasible after hospitalization. To minimize the risk of hospital-acquired colonization, enrollment and sampling occurred within 3 hours of admission in site 1 and later relaxed to 10 hours of admission in site 2. Sociodemographic and clinical information were collected using a pre-tested, semi-structured questionnaire. After quality checking and resolving any data collection errors, all data were entered into the electronic device (tablet) from the paper case report form (CRF) and synchronized with a central database weekly.

## Sample collection and laboratory procedures

Skin swab samples were collected from both axillae and groins using sterile transport swabs (HealthLink TransPorter by Copan Italia), placed in liquid Amies media, and transported to the icddr,b laboratory within six hours using standard procedures. Studies have suggested that though nares are the heavier source of colonization, the axilla (armpit) and groin are the most common and consistent sites of colonization of *C. auris* (14, 15). Additionally, we followed the CDC guidelines (https://www.cdc.gov/candida-auris/hcp/screening-hcp/c-auris-screening-patient-swab-collection-1.html) for selecting and collecting the skin swabs, using a composite swab from both sides of the axillae and groins for each individual. Samples were inoculated directly onto CHROMagar Candida Plus and incubated at 37°C for 48 hours. Presumptive *C. auris* colonies (beige, pink, white, or red) were identified by VITEK 2. One *C. auris* isolate, identified by VITEK 2, was re-tested by whole-genome sequencing (WGS).

## Sample size

Assuming a 2% *C. auris* colonization prevalence in the community, a 99% confidence interval, and a 2% margin of error, the minimum required sample size for detection was calculated as 327 using the formula, $n = [\log(1 - \text{Confidence})]/\log(1 - \text{Prevalence})$. As this was an exploratory study, we enrolled 400 participants from site 1. As no colonized cases were detected, after relaxing some inclusion criteria, we enrolled another 400 participants from site 2 to explore another community. The total number of participants was 800. Household members of any *C. auris*-colonized patients were subsequently planned to be enrolled to investigate potential transmission within households.

## Statistical analysis

Descriptive statistics were used to summarize sociodemographic, clinical, laboratory, and other relevant data. Medians with interquartile ranges (IQRs) were performed for skewed continuous variables. For comparison, the $\chi^2$ test was done for categorical variables, the

two-sample *t*-test for continuous variables, and the Mann–Whitney U test was done for median value. Data management and statistical analyses were performed using STATA SE (StataCorp LP, College Station, TX, 2017, version 15).

## RESULTS

During the study period, 5,994 admitted patients were screened, of whom 855 were eligible. After excluding 55 patients who refused to participate, 800 participants were enrolled. However, no *C. auris* was detected from any of the participants in our study (Fig. 1).

Most skin swabs were collected within one hour of hospitalization (median 0.53 hours, IQR: 0.30–0.97 hours; maximum: 8.6 hours). The majority of participants were enrolled from medicine wards (328, 41.0%), followed by pediatrics (223, 27.9%), gynecology and obstetrics (205, 25.6%), and surgery wards (44, 5.5%). Among the enrolled participants, 536 (67.0%) were adults (≥18 years), and females were more prevalent (500, 62.5%).

In the three months prior to admission, one-fourth (196, 24.5%) of participants had taken at least one antibiotic, and 21 (2.6%) had taken at least one antifungal. More than half of the participants (474, 59.3%) reported using treated water and having improved sanitation, which is sanitary latrine (677, 84.6%).

The sociodemographic status of the participants was nearly comparable across study sites for most variables. However, significant differences were found in the case of study wards, occupation, presence of chronic disease(s), sanitation facility, hospital visit(s) within the last 6 months, and the time difference between admission and sample collection (Table 1).

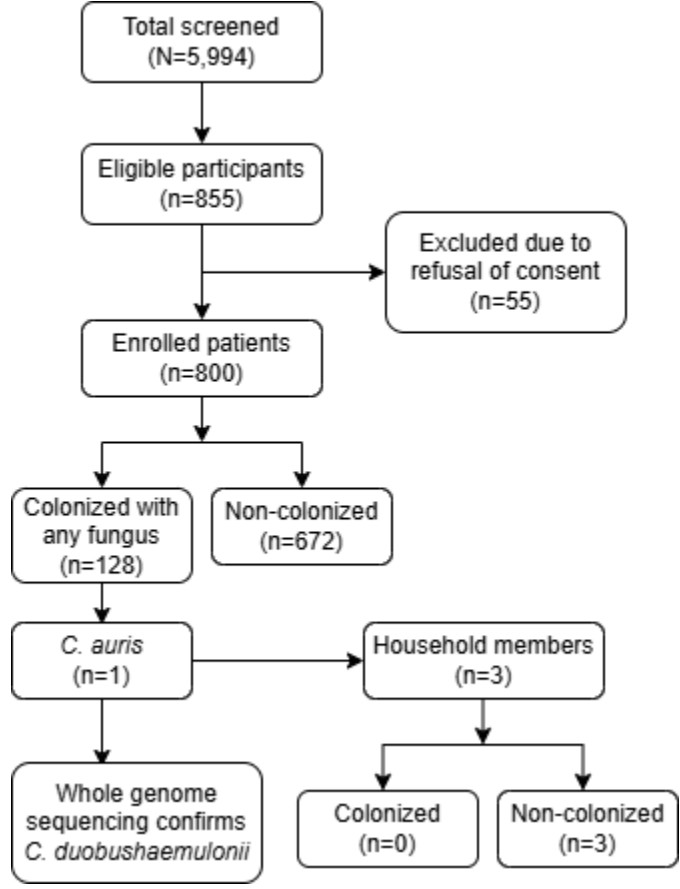

**FIG 1** Study flow diagram.

**TABLE 1** Sociodemographic characteristics of the enrolled patients by sites

| Characteristic | Site 1 (N = 400) n (%) | Site 2 (N = 400) n (%) | P value |
|---|---|---|---|
| Age in years, median (IQR) | 27 (12–45) | 24 (7.5–38.5) | 0.003 |
| Sex (female) | 245 (61.3) | 255 (63.8) | 0.47 |
| Study wards | | | |
| Medicine | 176 (44.0) | 152 (38.0) | 0.085 |
| Pediatrics | 101 (25.3) | 122 (30.5) | 0.098 |
| Gynae and obstetrics | 79 (19.8) | 126 (31.5) | <0.001 |
| Surgery | 44 (11.0) | 0 (0) | –[a] |
| Outside home occupation | 151 (37.8) | 118 (29.5) | 0.014 |
| Large family (>4 members) | 169 (42.3) | 162 (40.5) | 0.615 |
| Having chronic disease(s) | 111 (27.8) | 164 (41.0) | <0.001 |
| History of antifungal use within the last three months | 6 (1.5) | 15 (3.8) | 0.047 |
| History of antibiotic use within the last three months | 87 (21.8) | 109 (27.5) | 0.071 |
| Treated drinking water | 225 (56.3) | 249 (62.3) | 0.084 |
| Improved sanitation | 354 (88.5) | 323 (80.8) | 0.002 |
| Shared toilet (with other household) | 191 (47.8) | 186 (46.5) | 0.723 |
| Any healthcare visit within last six months | | | |
| No | 142 (35.5) | 138 (34.5) | 0.767 |
| <5 times | 230 (57.5) | 188 (47.0) | 0.003 |
| ≥5 times | 28 (7.0) | 74 (18.5) | <0.001 |
| Time difference between admission and sample collection (in hours), median (IQR) | 0.4 (0.3–0.8) | 0.7 (0.3–1.3) | <0.001 |

[a]–, not applicable.

VITEK 2 initially identified *C. auris* from the skin swab of one patient, collected within 0.6 hours of admission. However, whole-genome sequencing subsequently confirmed the isolate as *Candida duobushaemulonii* (Fig. 1). Skin swabs collected from three household members of this patient were negative for *C. auris*.

Overall, 128 (16%) participants were colonized with 136 fungal isolates. *Candida parapsilosis* was the most prevalent species (58, 7.3%), followed by *Candida albicans* (37, 4.6%). Eight participants (1.0%) were colonized with multiple fungal species. According to the CDC guidelines (https://www.cdc.gov/candida-auris/hcp/laboratories/identification-of-c-auris.html) and as evidenced by a multicenter study in Canada (16), we re-tested the two isolates of *Candida famata* and one isolate of *Candida lusitaniae* in VITEK MS, and none of them were detected as *C. auris*. The distribution of the 13 identified fungal species is shown in Fig. 2.

## DISCUSSION

This study investigated the burden of community-acquired *C. auris* colonization among newly hospitalized patients in Dhaka, Bangladesh, who had no recent healthcare exposure. While numerous studies have explored *C. auris* colonization in critically ill patients (3, 12), research specifically targeting community-acquired colonization remains limited. This investigation addresses this critical knowledge gap, given the increasing global concern regarding *C. auris* as a multidrug-resistant pathogen.

The key finding from this study was the absence of *C. auris* colonization among the study participants. Although VITEK 2 initially suggested *C. auris* in one skin swab sample, whole-genome sequencing definitively identified it as *Candida duobushaemulonii*. This underscores the necessity of confirmatory testing using MALDI-TOF or molecular methods, such as PCR or whole-genome sequencing, to avoid misidentification by phenotypic methods like VITEK 2. The absence of *C. auris* in household contacts of the misidentified individual further validated the accuracy of molecular identification.

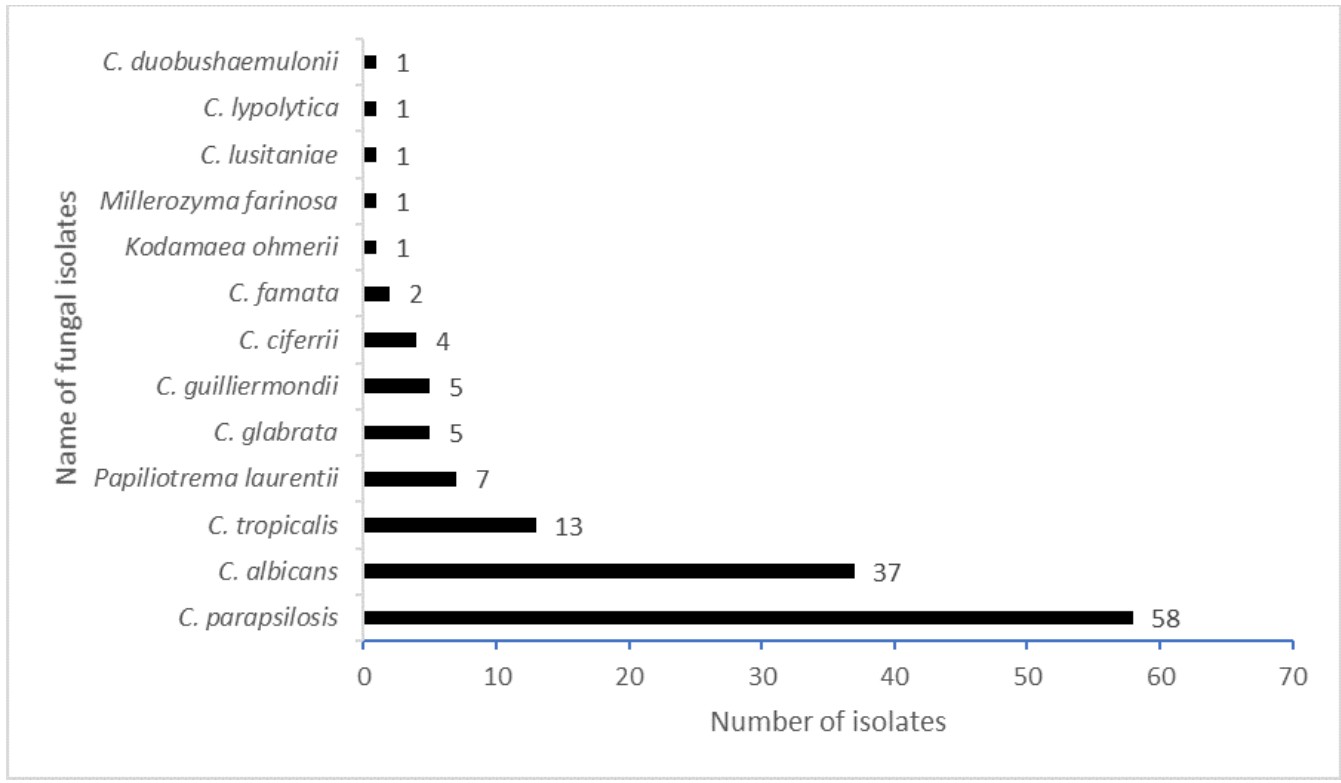

**FIG 2** Distribution of fungal species colonizing the skin of study participants.

This absence of *C. auris* contrasts with studies of asymptomatic carriers, which included patients with significant healthcare exposures and identified prior healthcare exposures and conditions such as mechanical ventilation within the preceding 90 days as risk factors (17). Eyre et al. detected *C. auris* on reusable hospital equipment but rarely in the non-healthcare environment (12), further suggesting healthcare-associated transmission. Rossow et al. identified prior acute care hospital visits as a risk factor for *C. auris* colonization among hospitalized patients (8). Although they also reported prior carbapenem and fluconazole use as risk factors for colonization (8), we did not detect *C. auris* colonization among the participants who reported antibiotic or antifungal use in the past three months prior to enrollment. To specifically investigate community acquisition, we implemented strict inclusion criteria by excluding individuals with any healthcare exposure within the past three months.

Our findings do not show evidence of community transmission of *C. auris* in the studied population, although active surveillance remains crucial. The study methodology, including enrollment within 10 hours of admission and axillary and groin sampling (as recommended for *C. auris* screening), was designed to detect *C. auris* colonization with a high likelihood of community origin.

Differences in some sociodemographic characteristics between the participants from two different sites did not affect the study results. Rather, these differences also indicated that despite differences among the study participants in both sites, no *C. auris*-colonized case was found. A significantly longer duration between admission and sample collection was due to a relaxed time window at one site, as mentioned in Materials and Methods.

We found a 16% prevalence of colonization with other fungal species upon admission, likely reflecting community-acquired colonization. This is lower than the 40% *Candida* colonization rate reported in a systematic review of ICU patients with sepsis (3) and also lower than colonization rates reported among healthy individuals (up to 60%) in different body sites, predominantly the gastrointestinal tract, genitourinary tract, and

skin (18, 19). However, since we only tested samples from the axilla and groin, areas not commonly associated with colonization by *Candida* species other than *C. auris* (14), this might be the reason behind the low prevalence rate of other *Candida* species found in our study. The most prevalent *Candida* species identified in our study were *C. parapsilosis*, *C. albicans*, and *C. tropicalis,* which are commonly part of the human microbiome (skin, mucous membranes, female genital tract, and gastrointestinal tract) and are generally considered commensal organisms in healthy individuals (20, 21). On the contrary, the most common and consistent sites of colonization for *C. auris* are the axilla and groin. However, the burden in nares, if found colonized, is quantitatively higher than the axilla or groin colonization (14). This prevalence of other *Candida* species, in the absence of *C. auris,* may also suggest that *C. auris* might require specific environmental conditions, such as healthcare facilities, which were not present in the studied community, or face competition from other *Candida* species.

This study provides valuable baseline data on fungal skin colonization in a community setting in Dhaka, Bangladesh. While the absence of *C. auris* is reassuring, continued surveillance remains essential, especially in hospital settings, as our earlier studies have detected both colonized and infected cases of *C. auris* among critically ill hospitalized patients (Chowdhury et al., unpublished). Future research should focus on longitudinal studies to monitor *C. auris* trends, population-based surveys to better understand community *C. auris* colonization, environmental sampling to identify potential reservoirs, and molecular epidemiology to elucidate transmission dynamics. These efforts will inform targeted public health interventions to prevent the spread of this potentially dangerous pathogen both within hospital settings and in communities.

This study has several limitations. First, the exclusion of individuals with any healthcare exposure in the past three months, while intended to isolate community-acquired cases, may have inadvertently excluded individuals who were genuinely colonized in the community but had recent contact with the healthcare system for unrelated reasons (e.g., outpatient visits, diagnostic tests) or contact with recently exposed individuals. This could have led to an underestimation of community prevalence. Second, sampling in our study was limited to axillary and groin swabs, as these are the most common and consistent sites of colonization, as evidenced by several studies. However, as *C. auris* can also colonize rarely in some other sites (e.g., nares, ears, and other skin folds), the presence of colonization in these sites could not be excluded. Third, while whole-genome sequencing was used to confirm a single initially identified *C. auris* isolate (which turned out to be *C. duobushaemulonii*), the primary identification method was VITEK 2 following culture on CHROMagar Candida Plus. While these are commonly used methods in clinical labs, they have the potential to misidentify *C. auris*. Nowadays, MALDI-TOF is considered a more accurate and rapid method for yeast identification, especially for distinguishing closely related *Candida* species. The use of VITEK 2 might have led to some misidentifications, although the WGS addressed the most critical one. Fourth, the study focused on newly hospitalized patients. This design does not capture *C. auris* colonization in the general community population who are not seeking hospital care. Therefore, it cannot provide a comprehensive picture of community prevalence. Fifth, the plan to enroll household members was contingent on finding *C. auris* colonization in the initial survey among hospitalized patients. Since no *C. auris* was detected (excluding one misidentified isolate), this aspect of the study was not implemented, limiting the ability to investigate household transmission. Moreover, the cross-sectional design limits the ability to establish causality and track changes in colonization over time. Finally, the sample size calculation was based on an assumed 2% prevalence. If the true prevalence in the community is even lower, the study may not have had sufficient power to detect it. Also, this study was conducted based on the findings from an earlier study where *C. auris* was found among critically ill patients in this region. Due to budget constraints, the study's location was only in Dhaka, Bangladesh, which limits the generalizability to other regions or countries.

## Conclusion

The absence of *C. auris* colonization among newly hospitalized study participants without recent healthcare exposure suggests limited community transmission in this setting and further supports the common understanding that transmission primarily occurs within healthcare settings. However, this finding should be interpreted in light of this study's limitations, including the strict inclusion criteria and limited sampling sites. Future prospective research should investigate *C. auris* colonization among individuals with risk factors, such as a recent history of hospitalization (especially in critical care units) or healthcare exposure as visitors, as well as their household members, across several study regions throughout the country. Also, patients who were non-colonized during their hospital stay can be followed up to check whether they become colonized later after discharge or if the colonization spreads among their household members, to better understand the potential for community spread and incubation periods. Additionally, robust infection prevention and control measures in hospitals remain essential to mitigate the spread of *C. auris* among admitted patients and other vulnerable populations.

### ACKNOWLEDGMENTS

We acknowledge with gratitude the commitment of the US CDC for funding this study and providing technical support. icddr,b also gratefully acknowledges our core donors for their support and commitment to its research efforts. Current donors providing unrestricted support include the Governments of Bangladesh and Canada.

We also thank all study participants and study hospital staff for their cordial support.

The findings and conclusions of this report are those of the authors and do not necessarily represent the official position of the Centers for Disease Control and Prevention (CDC).

Conceptualization: F.C., S.M., T.A.S., G.M.S.M. Data curation: M.A.I., K.A., K.H. Formal analysis: G.M.S.M., M.A.I. Funding acquisition: F.C. Investigation: D.A., D.S. Methodology: F.C., S.M., T.A.S., S.K., G.M.S.M. Project administration: F.C., G.M.S.M., S.K. Resources: M.N.I., A.R., M.M., M.M.R., F.N.M., M.S., R.Y., T.N.K., M.K., F.S., S.S.A. Software: K.H., M.A.I. Supervision: F.C. Validation: F.C. Visualization: M.A.I, K.H., G.M.S.M., F.C. Writing – original draft: G.M.S.M., F.C. Writing – review & editing: All authors of this manuscript.

### AUTHOR AFFILIATIONS

[1]International Centre for Diarrhoeal Disease Research, Bangladesh (icddr,b), Dhaka, Bangladesh

[2]University of New South Wales, Sydney, New South Wales, Australia

[3]Directorate General of Health Services, Ministry of Health and Family Welfare, Dhaka, Bangladesh

[4]Mugda Medical College Hospital, Dhaka, Bangladesh

[5]Shaheed Suhrawardy Medical College Hospital, Dhaka, Bangladesh

[6]Global Health Development/EMPHNET (Eastern Mediterranean Public Health Network), Dhaka, Bangladesh

### AUTHOR ORCIDs

Gazi Md. Salahuddin Mamun http://orcid.org/0000-0003-0907-7875

Tanzir Ahmed Shuvo http://orcid.org/0000-0003-4494-7856

Md. Aminul Islam http://orcid.org/0000-0002-9729-5233

### FUNDING

| Funder | Grant(s) | Author(s) |
| --- | --- | --- |
| Centers for Disease Control and Prevention | GH002259 | Fahmida Chowdhury |

## AUTHOR CONTRIBUTIONS

Gazi Md. Salahuddin Mamun, Conceptualization, Formal analysis, Methodology, Project administration, Visualization, Writing – original draft, Writing – review and editing | Tanzir Ahmed Shuvo, Conceptualization, Methodology, Writing – review and editing | Sanzida Khan, Methodology, Project administration, Writing – review and editing | Syeda Mah-E-Muneer, Conceptualization, Methodology, Writing – review and editing | Md. Aminul Islam, Data curation, Formal analysis, Software, Visualization, Writing – review and editing | Dilruba Ahmed, Investigation, Validation, Writing – review and editing | Kabid Ahmed, Data curation, Writing – review and editing | Debashis Sen, Investigation, Writing – review and editing | Kamal Hossain, Data curation, Software, Visualization, Writing – review and editing | Md. Nazmul Islam, Resources, Writing – review and editing | Aninda Rahman, Resources, Writing – review and editing | Mohammad Monir-Uz-Zaman, Resources, Writing – review and editing | Md. Mustafizur Rahman, Resources, Writing – review and editing | Fahmida Naz Mustafa, Resources, Writing – review and editing | Md. Salim, Resources, Writing – review and editing | Rubina Yasmin, Resources, Writing – review and editing | Md. Shafiur Rahman, Resources, Writing – review and editing | Tarak Nath Kundu, Resources, Writing – review and editing | Mostafa Kamal, Resources, Writing – review and editing | Farzana Sohael, Resources, Writing – review and editing | Sakina Shab Afroz, Resources, Writing – review and editing | Mahmudur Rahman, Writing – review and editing | Fahmida Chowdhury, Conceptualization, Funding acquisition, Methodology, Project administration, Supervision, Validation, Visualization, Writing – original draft, Writing – review and editing

## DATA AVAILABILITY

Data are available upon request and after obtaining ethical approval from the IRB of icddr,b. For data requests, kindly contact Ms. Shiblee Sayeed, Senior Manager, Research Administration, icddr,b, at shiblee_s@icddrb.org.

## ETHICS APPROVAL

The study protocol (PR-22159) was reviewed and approved by the Institutional Review Board (IRB) of the International Centre for Diarrhoeal Disease Research, Bangladesh (icddr,b). This IRB is comprised of two separate committees known as the Research Review Committee (RRC) and the Ethical Review Committee (ERC). The Centers for Disease Control and Prevention (CDC) was determined to be non-engaged in research, providing technical support, and thus, CDC IRB review was not required.

## ADDITIONAL FILES

The following material is available online.

### Open Peer Review

**PEER REVIEW HISTORY (review-history.pdf).** An accounting of the reviewer comments and feedback.

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
