## [Reviewer comments · Microbiology Spectrum]

Microbiology Spectrum

Absence of Community-acquired *Candida auris* Colonization among Newly Hospitalized Participants without Recent Healthcare Exposure from a Cross-Sectional Study in Dhaka, Bangladesh

Gazi Md. Salahuddin Mamun, Tanzir Shuvo, Sanzida Khan, Syeda Mah-E-Muneer, Aminul Islam, Dilruba Ahmed, Kabid Ahmed, Debashis Sen, Kamal Hossain, Md. Nazmul Islam, Aninda Rahman, Mohammad Monir-Uz-Zaman, Md. Mustafizur Rahman, Fahmida Mustafa, Md. Salim, Rubina Yasmin, Md. Shafiur Rahman, Tarak Kundu, Mostafa Kamal, Farzana Sohael, Sakina Afroz, Mahmudur Rahman, and Fahmida Chowdhury

Corresponding Author(s): Tanzir Shuvo, International Centre for Diarrhoeal Disease Research Bangladesh

Review Timeline:

Submission Date:	February 9, 2025
Editorial Decision:	March 11, 2025
Revision Received:	March 24, 2025
Accepted:	April 7, 2025

Editor: Gregory Wiedman

Reviewer(s): Disclosure of reviewer identity is with reference to reviewer comments included in decision letter(s). The following individuals involved in review of your submission have agreed to reveal their identity: Milena Kordalewska (Reviewer #2)

Transaction Report:

DOI: <https://doi.org/10.1128/spectrum.00393-25>

Re: Spectrum00393-25 (Absence of Community-acquired *Candida auris* Colonization among Newly Hospitalized Participants without Recent Healthcare Exposure from a Cross-Sectional Study in Dhaka, Bangladesh)

Dear Dr. Tanzir Ahmed Shuvo:

Thank you for the privilege of reviewing your work. Below you will find my comments, instructions from the Spectrum editorial office, and the reviewer comments.

Revision Guidelines

Sincerely,
Gregory Wiedman
Editor
Microbiology Spectrum

Reviewer #1 (Comments for the Author):

The article by Gazi et al describes the absence of community acquired *C. auris* infections in newly hospitalized patients in Bangladesh. *Candida auris* is a multidrug resistance human pathogen and is well known to be nosocomial. The authors in this study found no evidence of community acquired *C. auris* infections in newly admitted patients without recent healthcare exposure. The major issue of the article is that the authors did not properly clarify the importance of the study. Further the study

has several limitations many of which, the authors themselves have described. The study was performed in only two hospitals in Dhaka, Bangladesh. The authors should enlarge their study covering several regions of the country as well as collect samples from different sites of patients where *C. auris* are known to colonize.

Reviewer #2 (Comments for the Author):

The authors of Spectrum00393-25 manuscript aimed to determine the burden of community-acquired *Candida auris* colonization in patients without recent healthcare exposure. The manuscript is well written, but some corrections and clarifications are needed. Please see my comments below.

MAJOR COMMENTS

1. The manuscript would benefit from adding an epidemiological context of the study setting. What prompted you to run the study in this setting? Have *Candida auris* colonization or infection cases been reported in the captured geographic area? The only mention of colonization cases in Dhaka is in lines 93-94, but the details are scant.
2. VITEK 2 resulted in two and one isolates identified as *Candida famata* and *Candida lusitanae*, respectively (Figure 2). The ID of these isolates should be re-confirmed with another method since there have been reports of *C. auris* being misidentified as *C. lusitanae* and *C. famata* on VITEK 2. Please see the US CDC website for reference <https://www.cdc.gov/candida-auris/hcp/laboratories/identification-of-c-auris.html>
Also see: Ambaraghassi et al. Identification of *Candida auris* by Use of the Updated Vitek 2 Yeast Identification System, Version 8.01: a Multilaboratory Evaluation Study. *J Clin Microbiol.* 2019 Oct 23;57(11):e00884-19. doi: 10.1128/JCM.00884-19. PMID: 31413079; PMCID: PMC6812989.
3. Lines 230-233: Comment on the differences in the colonization prevalence in your study vs. published literature.
4. Axillae and groin have been determined to be the best body sites to detect *C. auris* colonization. Please revise lines 237-240 and 255-257 accordingly with the support of the published literature. Also see the CDC website: <https://www.cdc.gov/candida-auris/hcp/screening-hcp/index.html>

MINOR COMMENTS:

1. Have you thought of expanding your study and screening the enrolled patients at later timepoints of their hospital stay = look for colonization status change?

Response to Reviewers

Reviewer #1 (Comments for the Author):

The article by Gazi et al describes the absence of community acquired *C. auris* infections in newly hospitalized patients in Bangladesh. *Candida auris* is a multidrug resistance human pathogen and is well known to be nosocomial. The authors in this study found no evidence of community acquired *C. auris* infections in newly admitted patients without recent healthcare exposure. The major issue of the article is that the authors did not properly clarify the importance of the study. Further the study has several limitations many of which, the authors themselves have described. The study was performed in only two hospitals in Dhaka, Bangladesh. The authors should enlarge their study covering several regions of the country as well as collect samples from different sites of patients where *C. auris* are known to colonize.

Authors' response: Thank you for your valuable time to review our article. We've addressed your comments that has improved the quality of it.

We've updated the importance of this study as below and hope that it is now properly clarified:

Candida auris, a rapidly emerging multidrug-resistant fungal pathogen with high mortality, posing a critical global health threat, particularly in healthcare settings. While extensive research has focused on its colonization among hospitalized patients, its potential presence and community transmission remains largely unexplored. An earlier study among critically ill patients in Bangladesh detected four colonized cases of *Candida auris* without hospitalization history within the past year and admitted directly from home (Fahmida C. et al., submitted for publication). Notably, three resided in the same district and one in an adjacent district, raising serious concerns about possible community spread. This study is crucial in addressing this knowledge gap by assessing the burden of community-acquired *Candida auris* colonization among newly hospitalized patients in Dhaka. Understanding its potential transmission outside hospital is vital for shaping public health responses, guiding infection control strategies, and strengthening global surveillance efforts to mitigate the spread of this highly-resistant pathogen. [Lines 57 – 69 of track change version].

We couldn't conduct this study beyond two sites due to budget constraints and now mentioned under limitations [Lines: 306 – 309 of track change version].

Due to similar reason and also as evidences showed that axilla and groin are the most common and consistent sites of *Candida auris* colonization, our study samples were limited to both axillae and groins. This is now added under Sample Collection and Laboratory Procedures section of Methods (Lines: 153 – 158 of track change version).

Reviewer #2 (Comments for the Author):

The authors of Spectrum00393-25 manuscript aimed to determine the burden of community-acquired *Candida auris* colonization in patients without recent healthcare exposure. The manuscript is well written, but some corrections and clarifications are needed. Please see my comments below.

Authors' response: Thank you for kindly reviewing our manuscript and providing your valuable feedback that helps to improve the quality of it. We've addressed the comments raised by you as follows.

MAJOR COMMENTS

1. The manuscript would benefit from adding an epidemiological context of the study setting. What prompted you to run the study in this setting? Have *Candida auris* colonization or infection cases been reported in the captured geographic area? The only mention of colonization cases in Dhaka is in lines 93-94, but the details are scant.

Authors' response: We thank the reviewer for this important suggestion. A recently completed study in Dhaka found 4% *Candida auris* colonization among ICU patients within 48 hours of admission and two cases of bloodstream infection. Among the colonized patients, four had no history of hospitalization within the past year and living in the nearby geographic area. So, due to the possibility of community transmission among them, we conducted this study to create a better evidence on it. We've added this information in lines 106 – 110 of track change version.

2. VITEK 2 resulted in two and one isolates identified as *Candida famata* and *Candida lusitanae*, respectively (Figure 2). The ID of these isolates should be re-confirmed with another method since there have been reports of *C. auris* being misidentified as *C. lusitanae* and *C. famata* on VITEK 2. Please see the US CDC website for reference <https://www.cdc.gov/candida-auris/hcp/laboratories/identification-of-c-auris.html>

Also see: Ambaraghasi et al. Identification of *Candida auris* by Use of the Updated Vitek 2 Yeast Identification System, Version 8.01: a Multilaboratory Evaluation Study. *J Clin Microbiol.* 2019 Oct 23;57(11):e00884-19. doi: 10.1128/JCM.00884-19. PMID: 31413079; PMCID: PMC6812989.

Authors' response: Thank you for your great suggestion. We've re-tested the 2 isolates of *Candida famata* and 1 isolate of *Candida lusitanae* in VITEK MS and none of them were detected as *Candida auris*. We've now mentioned this in Results section [Lines 211 – 215 of track change version].

3. Lines 230-233: Comment on the differences in the colonization prevalence in your study vs. published literature.

Authors' response: Thank you for providing us the opportunity to explain the difference as found in the colonization prevalence of other *Candida* species. Except *Candida auris*, other *Candida* species are mostly colonized in gastrointestinal and genitourinary tract. As our aim was to find out the burden of *C. auris*, so we didn't collect samples from the gastrointestinal and genitourinary tracts. This is the reason behind our low prevalence of other *Candida* species. We've now explained this in lines 257 – 261 of track change version.

4. Axillae and groin have been determined to be the best body sites to detect *C. auris* colonization. Please revise lines 237-240 and 255-257 accordingly with the support of the published literature. Also see the CDC website: <https://www.cdc.gov/candida-auris/hcp/screening-hcp/index.html>

Authors' response: We agree with the reviewer's comment. Studies have suggested that though nares are the heavier source of colonization, axilla (armpit) and groin are the most common and consistent sites of colonization of *Candida auris*. Additionally, we followed the CDC guidelines for selecting and collecting the skin swabs from both sides of the axillae and groins as a composite swab from each individual. We've now written this under Sample Collection and Laboratory Procedures section of Methods (Lines: 153 – 158 of track change version). Additionally, according to the suggestions, we've also revised the lines 264 – 269 and 286 – 290.

MINOR COMMENTS:

1. Have you thought of expanding your study and screening the enrolled patients at later timepoints of their hospital stay = look for colonization status change?

Authors' response: If budget becomes, available, then we can expand the study in other sites for the generalizability of this study. Also, future studies can be done to screen the high-risk patients such as discharged from ICU to see whether they can spread among their household members. We've now added this under Conclusion section [Lines 319 – 321 of track change version].

Re: Spectrum00393-25R1 (Absence of Community-acquired *Candida auris* Colonization among Newly Hospitalized Participants without Recent Healthcare Exposure from a Cross-Sectional Study in Dhaka, Bangladesh)

Dear Dr. Tanzir Ahmed Shuvo:

Your manuscript has been accepted, and I am forwarding it to the ASM production staff for publication. Your paper will first be checked to make sure all elements meet the technical requirements. ASM staff will contact you if anything needs to be revised before copyediting and production can begin. Otherwise, you will be notified when your proofs are ready to be viewed.

Sincerely,
Gregory Wiedman
Editor
Microbiology Spectrum

Reviewer #2 (Comments for the Author):

I am satisfied with the responses provided by the authors.